# Mortality of Invasive Pneumococcal Disease following Introduction of the 13-Valent Pneumococcal Conjugate Vaccine in Greenland

**DOI:** 10.3390/vaccines12020179

**Published:** 2024-02-09

**Authors:** Kristiana Alexandrova Nikolova, Mikael Andersson, Hans-Christian Slotved, Anders Koch

**Affiliations:** 1Department of Epidemiology Research, Statens Serum Institut, Artillerivej 5, DK-2300 Copenhagen, Denmark; aso@ssi.dk; 2Department of Bacteria, Parasites & Fungi, Statens Serum Institut, Artillerivej 5, DK-2300 Copenhagen, Denmark; hcs@ssi.dk; 3Department of Infectious Diseases, Rigshospitalet University Hospital, Blegdamsvej 9, DK-2100 Copenhagen, Denmark; ako@ssi.dk; 4Department of Infectious Disease Epidemiology & Prevention, Statens Serum Institut, Artillerivej 5, DK-2300 Copenhagen, Denmark; 5Greenland Center for Health Research, Institute of Health and Nature, Ilisimatusarfik (University of Greenland), Manutooq 1, P.O. Box 1061, Nuussuaq 3905, Greenland; 6Department of Internal Medicine, Queen Ingrids Hospital, Nuuk 3900, Greenland

**Keywords:** invasive pneumococcal disease, pneumococcal conjugate vaccine, Inuit, mortality, case fatality rate, mortality rate, standardized mortality ratio, pneumococcal serotypes, PCV13, IPD

## Abstract

Before the incorporation of the 13-valent pneumococcal conjugate vaccine (PCV13) into the childhood vaccination regimen in Greenland in 2010, Inuit populations experienced a substantial prevalence of invasive pneumococcal disease (IPD). The PCV13 introduction has been shown to markedly reduce the incidence of IPD. This current study estimated the impact of PCV13 introduction on IPD mortality in Greenland. This was a nationwide register-based study using all available data on IPD cases 1995–2020 in Greenland. Thirty-one-day IPD case fatality rates (CFR), and all-cause and mortality rates associated with IPD during the period before the introduction of PCV13 (January 1995 to September 2010) were compared with those observed in the post-PCV13 era (September 2010 to October 2020). Standardized mortality ratios (SMRs) expressed differences in mortality by sex, age, region, ethnicity, comorbidity, and serotype. IPD CFR decreased with 24.5% from the pre- to the post-PCV13 period. SMR in IPD patients decreased by 57% (95% CI, 36–75%), and a reduction occurred in all age groups. While SMR in IPD persons ≥60 years remained virtually unchanged, there were no IPD-related deaths in persons ≤39 years in the post-PCV13 period. In conclusion, IPD-related mortality has decreased in Greenland following PCV13 introduction in 2010 in the country.

## 1. Introduction

The Inuit populations of the Arctic regions of Alaska, Canada, and Greenland have historically suffered from high incidence rates of invasive pneumococcal disease (IPD) [1,2,3,4].

Surveillance of invasive pneumococcal disease (IPD) in 1986 showed that the Inuit population of Alaska had the highest reported average overall IPD rate in the world, which was four times higher than the rate of the non-native population [5]. Given the shared genetic, social, and environmental risk factors among Inuit populations in the North American Arctic, including common living conditions, socio-economic challenges, and levels of comorbidity, the International Circumpolar Surveillance System (ICS) for IPD was established. This surveillance system has confirmed high rates of IPD in other Inuit populations [5]. Yet, few studies have evaluated the burden of IPD in the Inuit population of Greenland [3,6,7,8].

The 13-valent pneumococcal conjugate vaccine Prevnar 13^®^ (PCV13; Pfizer, New York, NY, USA), covering 13 *Streptococcus pneumoniae* serotypes was introduced in the childhood vaccination program (CVP) by 1 September 2010, in Greenland [9]. The CVP is free and is eligible to all children with a permanent address in Greenland [10]. Before the introduction of the PCV13, IPD rates among Inuit in Greenland were up to four times higher than among non-Inuit in the country [6,8].

In our previous study [8], it was shown, that after the introduction of the PCV13, the overall incidence of IPD declined with 30% and IPD caused by vaccine serotype (VT) declined with 55%. Moreover, VT-IPD incidence rates among children ≤1 years declined with 30%, and no VT-IPD cases among children 2–4 years were observed post-PCV13 introduction. Besides this direct effect, the vaccine has also shown to have an indirect (herd) effect.

The indirect vaccine protection of persons older than the targeted group for vaccination has shown to be due to reduced carriage of *S. pneumoniae* in the vaccinated persons, thereby reducing the transmission of the infection [2]. However, as in many other countries, the introduction of the PCV13 in Greenland has been associated with shifts in nasopharyngeal pneumococcal serotype distribution with increased carriage of non-vaccine serotypes (NVT), increasing NVT-IPD incidence rates among adults ≥60 years [8].

Worldwide, it is estimated that 1.6 million people of all ages die each year from IPD, including about 814,000 deaths among children under the age of five [11]. IPD case-fatality rates (CFR) have been shown to be about four times higher in Inuit populations than in non-Inuit [4,6].

Outcomes of IPD in terms of severity and mortality are influenced by host characteristics such as age, socioeconomic status, ethnicity, alcoholism, immunodeficiency, and comorbidities [3,12,13]. Moreover, studies have shown an association between the outcome of IPD and particular serotypes [12,14,15].

While our previous study demonstrated the effect of PCV13 on serotype distribution in Greenland [8], little is known regarding the effect of pneumococcal conjugate vaccines (PCVs) on IPD-related mortality, particularly in Greenland.

The aim of this study was to describe changes in IPD-related mortality and fatality after the introduction of the PCV13 in 2010 in Greenland, and how this varies by age, sex, region, ethnicity, and pneumococcal serotype.

## 2. Materials and Methods

This study is a part of a nationwide register-based study of invasive pneumococcal disease following the introduction of PCV13 vaccination in the Greenlandic childhood vaccination program. A detailed description of data collection can be found in Nikolova et al. [8]. Briefly, IPD cases were identified from the Greenlandic National Inpatient Registry (NIR) and from laboratory records. Medical journals were then gone through in order to confirm the identified cases. Lastly, further IPD-cases were retrieved from case reports to the National Board of Health in Greenland.

### 2.1. Setting

This study involves the Greenlandic population. The total population of Greenland is about 56,000, whereof 18,800 live in the capital, Nuuk. Eighty-one percent of Greenland is covered by ice [16].

Health care is provided free of charge to all citizens in Greenland, including vaccinations in the childhood vaccination program [10,17]. The central and national referral hospital in Greenland, The Queen Ingrid’s Hospital (QIH), is located in Nuuk. The QIH operates as the only microbiological laboratory in Greenland, consequently handling all microbiological samples for culturing from across the country. Upon identification, *S. pneumoniae* cultures are sent for sero- and genotyping from the QIH to Statens Serum Institut in Copenhagen, Denmark [8].

About 850 persons are born and 500 persons die each year in Greenland. Life expectancy is shorter in Greenland than the average in the western world. In Greenland the average life expectancy is 68.3 years for men and 73.0 years for women. This comparably shorter life expectancy is primarily due to high mortality rates caused by accidents and suicide [16].

### 2.2. Study Population, Definitions, and Data Sources

Similar to our previous study did this study include all persons diagnosed with IPD, either clinically or microbiologically, or both, at any point of time between 1 January 1995 and 1 October 2020 in Greenland. A thorough description of the study population, and on definitions are to be found in Nikolova et al. [8].

Data on the size and mortality figures for the Greenlandic population spanning from 1995 to 2020 were obtained from Statistics Greenland, the central statistical organization in Greenland. The midyear estimates preceding each winter season served as the denominators. Information pertaining to sex, past and current place of residence, and ethnicity, defined by the patient’s and parents’ places of birth, was gathered for both the entire population denominator and all cases of IPD.

### 2.3. Statistical Methods and Measures of Mortality

If death occurred within 31 days after hospitalization, we defined it as IPD-related mortality. Patients were censored 31 days after hospitalization.

To assess the effects of the PCV13 vaccine following its implementation in Greenland in September 2010, mortality associated with IPD during the pre-PCV13 period (from 1 January 1995, through 31 August 2010) was compared with that in the post-PCV13 period (from 1 September 2010, through 1 October 2020). We used three measures for mortality: 1. IPD case-fatality rates (CFRs) were defined as the proportion of IPD patients who died within 31 days after hospitalization, overall and by serotype, over the entire study period and over the respective study periods, expressed in percentages. 2. Crude all-cause mortality rates were defined with the number of deaths in the Greenlandic population in a given year as nominator and the mid-year population number as denominator, expressed per 1000 person-years (PYRS). Crude IPD-related mortality rates were defined as number of deaths within 31 days for persons with IPD as nominator and 1000 PYRS of the Greenlandic population as denominator. In addition to crude mortality rates observed in the Greenlandic population, age-standardized rates for overall mortality and mortality related to IPD were calculated through the direct standardization method [18]. This involved applying a weighted average of age-specific mortality rates per 1000 PYRS, where the weights were determined by the proportions of individuals within the corresponding age groups of the World Health Organization (WHO) World Population Standard [19]. Weighing does not change PYRS or the age-specific mortality rates, but changes the balance between age groups and number of deaths. Use of the direct standardization method measure makes it possible to compare mortality rates in Greenland with those of the world population as the Greenlandic population’s age structure is changing and differs from that of the world population. 3. Standardized mortality ratios (SMR) were calculated as the ratios of observed deaths from IPD relative to expected IPD deaths given the death rates in the standard population in Greenland, overall and stratified by sex, age, region, serotype, ethnicity, and Charlson Comorbidity Index (CCI) scores [20]. SMR in IPD patients stratified by region was compared using ANOVAs on Poisson regression models with the observed number of cases as outcome and the logarithm of the expected number of cases as offset using functions glm and anova from the Stats package in R (R version 4.3.0).

Kaplan–Meier curves were used to illustrate 31-day mortality among patients living in Greenland. Differences in 31-day mortality were estimated through Log-Rank tests. Cox regression was used to investigate the effect of potential confounders including sex and CCI score, stratified by age and time-period. The proportional hazard assumption was evaluated using statistical tests and visual inspection of the scaled Schoenfeld residuals in R packages Survival and Survminer (R version 4.3.0).

We used multiple logistic regression models to determine the odds ratios (ORs) for IPD-related mortality by serotype, sex, clinical diagnose, ethnicity, and region. Differences in proportions were calculated using the chi-square test.

Statistical analyses were performed using R version 4.3.0.

A *p*-value < 5% was considered significant.

## 3. Results

Of the entire cohort of 295 IPD patients, 66 died within 31 days after IPD hospitalization representing a CFR of 22.4%. During the pre-PCV13 period (January 1995–September 2010), 49 out of 206 (CFR 23.8%) IPD patients died within 31 days after IPD hospitalization, while 17 out of 89 (CFR 19.1%) IPD patients died within 31 days after IPD hospitalization during the post-PCV13 period (September 2010–October 2020). This difference was not significant (*p* = 0.463).

The average length of hospitalization for all IPD cases was 14 days. This includes 15 cases who had a length of hospitalization more than 40 days. The average length of hospitalization was 14 days (range 0 to 121 days) for the pre-PCV13 period and 15 days (range 0 to 138 days) for the post-PCV13 period.

### 3.1. Mortality Rates

Table 1 shows crude all-cause mortality rates and age-standardized all-cause mortality rates for the entire population in Greenland and crude IPD-related mortality rates and age-standardized IPD-related mortality in Greenland.

Although there was a statistically significant change in crude all-cause mortality in Greenland from the pre- to the post-PCV13 period (9.1 per 1000 PYRS to 9.7 per 1000 PYRS) (*p* < 0.05), the mortality rates were similar. However, after age-standardization, all-cause mortality rates in Greenland decreased from pre- to the post-PCV13 period (16.4 per 1000 PYRS to 12.8 per 1000 PYRS), which was statistically significant (*p* < 0.05).

For the whole period, crude IPD-related mortality rate was up to 350-times higher than the crude all-cause mortality rate for the population in Greenland. This rate decreased from the pre- to the post-PCV13 period (3589 per 1000 PYRS to 2658 per 1000 PYRS), but after age-standardization, IPD-related mortality rates increased from the pre- to the post-PCV13 period (3347 per 1000 PYRS to 4441 per 1000 PYRS).

Table 2 shows SMRs in patients IPD. Overall, SMR in the pre-PCV13 period was significantly higher than in the post-PCV13 period (267.6 vs. 114.2) (*p* = 0.003). SMR in patients with IPD was in general higher during the pre-PCV13 period compared to the post-PCV13 period when IPD-related mortality cases were stratified by sex, *S. pneumoniae* serotypes, age group, region of Greenland, ethnicity, and level of Charlson Comorbidity Score. There was no IPD-related mortality in patients up to 39 years of age during the post-PCV13 period.

IPD patients had a non-significant higher 31-day survival probability during the post-PCV13 period, compared to the pre-PCV13 period (*p* = 0.31) (Figure 1a). The difference between the pre-PCV13 period and the post-PCV13 period for 31-day IPD survival probability was greater when adjusting for sex, age, and CCI score (*p* = 0.08) (Figure 1b).

Figure 2 shows unadjusted IPD-related mortality odds ratios (OR) for the pre- and the post-PCV13 period. Except for IPD patients from the east and the west region of Greenland during the pre-PCV13 period, IPD patients outside of the capital Nuuk during both the pre- and the post-PCV13 period were associated with higher 30-day IPD-related mortality compared to IPD patients in Nuuk. However, only the association of 31-day IPD-related mortality between the west and east regions of Greenland during the post-PCV13 period and Nuuk was statistically significant.

IPD patients with meningitis, compared to IPD patients with non-meningitis, had a higher 31-day IPD-related mortality both during the pre- and the post-PCV13 period, but this was only statistically significant for the post-PCV13 period.

Patients with VT-IPD had a lower 31-day IPD-related mortality during both periods compared to patients with NVT-IPD.

There were no IPD-related deaths in non-Inuits during the post-PCV13 period. Inuits had almost the same 31-day IPD-related mortality during the pre-PCV13 period as non-Inuits. Females had a lower 31-day IPD-related mortality compared to males during both the pre- and the post-PCV13 periods.

### 3.2. Mortality by Serotypes

Of the total 295 IPD patients in the entire study period, only 185 IPD patients had their isolates serotyped. Of these 185 isolates, 115 (55.8%) were serotyped during the pre-PCV13 period and 70 (78.6%) were serotyped during the post-PCV13 period.

Figure 3 shows the proportion of deaths by serotype. Of the 185 serotyped isolates, VT serotypes 3, 4, 6A, 7F, 9V, 14, and 18C all caused IPD-related deaths during the pre-PCV13 period. Only VT serotype 4 and 19F caused IPD-related deaths during the post-PCV13 period (Figure 3: left panel). During the pre-PCV13 period, the NVT serotypes that caused IPD-related deaths, were serotypes 8, 12F, 15B, and 22F. During the post-PCV13 period, NVT serotypes 9N, 10A, 10B, 16F, 20, and 22F caused IPD-related deaths (Figure 3: right panel).

## 4. Discussion

In this register-based study of all IPD cases in Greenland 1995–2020, we found that CFR for IPD decreased with 24.5% from the pre- (1995–2010) to the post-PCV13 (2010–2020) period. The SMR decreased significantly in IPD patients from the pre- to the post-PCV13 period (0.43, 95% CI 0.25–0.75), regardless of sex, ethnicity, comorbidity, and age. More so, there was no IPD-related death in patients up to 39 years of age in the post-PCV13 period.

For IPD patients in the post-PCV13 period, 31-day survival probability was higher, but not significant, compared to IPD patients in the pre-PCV13 period, also after adjusting for sex, age, and CCI score. In addition, there was no difference in length of hospital stay between the two periods.

During the years of 1999–2020, life expectancy in Greenland increased from 63 years to 69 years for men and from 69 years to 73 years for women [21]. This may explain the increase in crude all-cause mortality rate in Greenland from the pre- to the post-PCV13 period. However, despite the increasing life expectancy in Greenland, it is still below the estimated global average life expectancy of around 73 years [22]. As the population in Greenland changes towards becoming older [16], we used age-standardization adjustment to see how the rate would change if the age structure of the Greenlandic population was the same in both periods and as that of the estimated world population. After age-standardization, we observed a decrease in crude all-cause mortality in the post-PCV13 period in Greenland.

IPD is a disease that carries a significant mortality in Greenland, which is reflected by the 350-times higher risk of IPD-related mortality when compared with all-cause mortality. However, crude IPD-related mortality rates decreased from the pre- to the post-PCV13 period with 26% (*p* = 0.291). On that note, the reason for the 33% (*p* = 0.232) increase in IPD-related mortality after age-standardization from the pre- to the post-PCV13 period is because the elderly IPD cases in the post-PCV13 period are very few but have a high mortality rate, affecting the world-weighted deaths.

It is well known that age influences the outcome of IPD. In this study, SMR in patients with IPD decreased in all age groups from the pre- to the post-PCV13 period. In fact, no children with IPD under the age of 5 years or persons in the age group 5–39 died within 31 days after IPD diagnosis in the post-PCV13 period. These observations are in line with studies from other countries, where PCV13 introduction has been associated with a reduction in IPD-related mortality rates in children aged <5 years [23,24,25,26] in young adults, and in adults [24,27].

Previous studies from Greenland and other countries have shown that mortality from IPD is greater in the elderly ≥65 years than in any other age group [3,6,12,28,29,30]. In our study, SMR was lowest in patients with IPD older than 60 years during the entire study period with almost similar SMRs for this age group in the two periods. Similar results were observed in a study from Spain, where no significant difference in IPD-related mortality rates in patients ≥65 years was observed [27]. This is, however, not the general case, as some studies have observed a reduction in IPD-related mortality rates and CFRs in the elderly [24,25], although the decrease in IPD-related mortality rate in persons aged ≥ 65 years observed in one study was minimal [25].

CFRs of both VT- and NVT-IPD decreased among adults aged 40–59 years from the pre- to the post-PCV13 period (19.4–16.6% and 30–18.8%, respectively). However, in adults ≥60 years CFRs of VT-IPD remained the same (20% in both periods), and CFR of NVT-IPD in the same age group increased from 12.5% in the pre-PCV13 period to 25.9% (*p* = 0.753) in the post-PCV13 period. Increases in death from NVT-IPD among the elderly has also been observed in the study from Spain, where a trend towards an increase in NVT-IPD-related mortality rates was observed in patients ≥65 years following PCV13 introduction [27].

The above findings suggest that while PCV13 has a direct effect on mortality of the vaccinated persons, and a herd protection of the adults, there is no effect of PCV13 introduction for children on mortality of the elderly above 60 years of age.

Pneumococcal serotypes differ in invasiveness [31,32]. It has been suggested that serotypes that have a high risk of causing invasive disease infect healthy individuals and therefore behave as ‘primary pathogens’, whereas serotypes with a lower potential to cause invasive disease cause disease in patients with underlying conditions, and in those patients cause more severe disease and mortality thus behaving as ‘opportunistic pathogens’ [31]. In particular, serotypes 3, 6A, and/or 19F are among the serotypes described as less invasive and thus associated with the highest mortality rates in persons with comorbidities [13,30,33,34]. In our study, persons who died from a suggested less invasive serotype had low levels of comorbidity, except for one person who had a moderate level of comorbidity as the highest level. Moreover, persons with “high-invasive serotypes” (4, 7F, 14, and 18C) had overall low levels of comorbidity, and there were more deaths from these serotypes than from the suggested less-invasive serotypes. Thus, our study could not support an association between potential suggested serotype-invasiveness and mortality, which is in line with one other study [12].

Compared with non-Inuits, Inuits had a higher IPD-related SMR. This is in accordance with previous studies of IPD in Greenland [3,6]. Deaths among non-Inuits were only observed in the pre-PCV13 period, but as the number of IPD-related deaths among this group was very small (*n* < 5), there may not be a true difference between the two periods for this group.

We found a decrease in SMR in patients with IPD from the pre-PCV13 to the post-PCV13 period in all levels of comorbidity defined by CCI-score level groups. The decrease in SMR was less pronounced in IPD patients with high CCI-scores confirming that underlying disease are effect modifiers of IPD-related mortality as observed in other studies [12,13,23,30]. Nevertheless, our observations support that PCV13 has protective effects against IPD-related mortality in all levels of comorbidity.

There were clear differences in SMR overall, and over time, between the regions of Greenland; North and South regions of Greenland had the highest SMR of IPD in the pre-PCV13 period which decreased in the post-PCV13 period, while IPD SMR in the East and West regions were lowest during the pre-PCV13 period but increased in the post-PCV13 period. The highest decrease in IPD SMR from the pre- to the post-PCV13 period was, however, observed in Nuuk (0.17, 95% CI: 0.06–0.44). Reasons for these opposite results between regions are unclear, but as diagnostic and treatment possibilities differ markedly between the regions (mainly between Nuuk and coastal hospitals) we believe that such differences over time and place may explain the observations.

IPD patients with meningitis compared to IPD patients with non-meningitis were associated with the highest mortality rates in both periods. Studies from other countries have shown decreases in fatal cases of non-meningitis IPD presentations, such as bacteremia and/or septicemia, after PCV introduction [23,25], yet most of the IPD-related deaths remained due to meningitis [23].

Thus, our study shows that IPD-related mortality has decreased in Greenland since the introduction of the PCV13 in 2010 in terms of numbers who die from the disease. Decreases in IPD-related mortality after PCV13 implementation have generally been observed in studies from other countries [24,25,27]. Although we did not have access to information of vaccine status in IPD cases, we believe from our previous [8] and current figures that it is likely that the reduction in IPD-related mortality following vaccine introduction is caused both by a reduction in IPD incidence and a lower risk of death.

Obviously, a temporal association between PCV13 introduction and reduced mortality may not incur causality. Other factors may also account for the reduced mortality, such as better access to health facilities and to laboratory diagnostic testing, and more timely hospital treatment. The finding of a markedly reduced mortality from IPD in the capital Nuuk, where the central hospital in Greenland is located, compared to other regions, could suggest that improved medical treatment could play a role in the reduced mortality following PCV13 introduction.

The strengths of this study are that our population was well-defined, and the study period was long, enabling us to identify long-term association between PCV13 introduction and mortality. We were able to obtain exact demographic information of the study population as we could uniquely identify case patients on a personal identifiable level. Moreover, we were able to obtain a full background population size.

There are a number of limitations in this study. First, the number of observed deaths is small and therefore estimates of, e.g., rates may be interpreted with care. Second, a temporal association between PCV13 introduction and reduction in mortality may not reflect causality as discussed above. Third, IPD-related mortality was defined as deaths within 31 days after hospital admission for IPD, irrespectively of the actual cause of death as we did not have access to information of causes of death for the majority of patients. Fourth, by censoring 31 days after IPD hospitalization we exclude potential cases that might have died from IPD beyond 31 days after hospitalization. Finally, information on serotypes was incomplete for the majority of our study population, and the sample sizes for each specific serotype was small.

## 5. Conclusions

In conclusion, IPD-related mortality has decreased in Greenland following PCV13 introduction in 2010 in the country. While a number of factors may account for this, we observed a reduction both in the absolute number of IPD deaths in Greenland and in severity of IPD cases in terms of a lower CFR after PCV13 introduction. In the post-PCV13 period, there were no IPD-related deaths in children and adults up to 39 years of age, indicating a direct and indirect benefit of PCV13 on IPD mortality. However, this was not observed in persons aged ≥ 60 years. Thus, while PCV13 introduction may have attributed to a reduced mortality from IPD in the younger age groups, with an increasing life expectancy in Greenland and changes in serotype distribution in the elderly, IPD remains a public health problem in Greenland.

## Figures and Tables

**Figure 1 vaccines-12-00179-f001:**
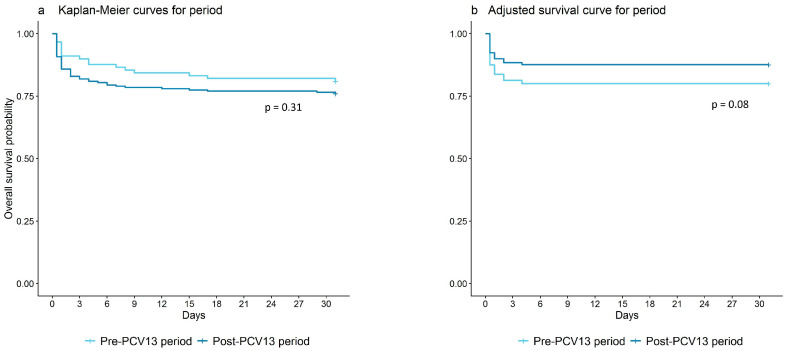
Comparison of 31-day survival probability from first day of invasive pneumococcal disease (IPD) hospitalization according to pre (1995–2010) (light blue line) and post (2010–2020) (dark blue line) introduction of the 13-valent pneumococcal conjugate vaccine (PCV13) with or without adjustment for potential confounders. (**a**) unadjusted Kaplan–Meier survival curve for pre (1995–2010) (light blue line) and post (2010–2020) (dark blue line) introduction of the PCV13. (**b**) survival curve adjusted for sex, age, and CCI score pre (1995–2010) (light blue line) and post (2010–2020) (dark blue line) introduction of the PCV13 using a Cox-regression.

**Figure 2 vaccines-12-00179-f002:**
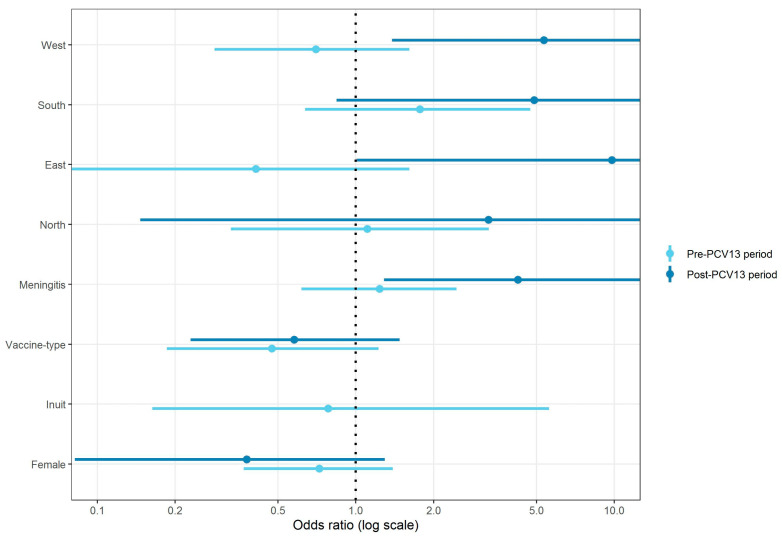
Multivariable logistic regression analysis of 31-day invasive pneumococcal disease (IPD) related mortality by pre (1995–2010) and post (2010–2020) introduction of the 13-valent pneumococcal conjugate vaccine (PCV13). Vertical black dotted line at odds ratio = 1. Unadjusted odds ratios (ORs) for region of Greenland (Nuuk as reference), sex (male as reference), clinical diagnose (non-meningitis as reference), ethnicity, and serotype (non-vaccine serotype (NVT) as reference).

**Figure 3 vaccines-12-00179-f003:**
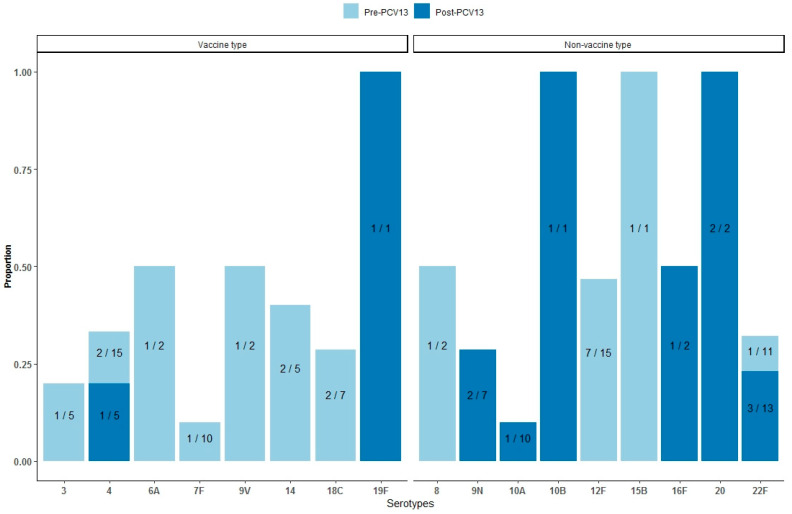
Proportion of invasive pneumococcal disease (IPD) deaths by vaccine serotypes (VT) and non-vaccine serotypes (NVT), and by pre- (1995–2010) and post- (2010–2020) introduction of the 13-valent pneumococcal conjugate vaccine (PCV13). Abbreviations: PCV13: 13-valent pneumococcal conjugate vaccine.

**Table 1 vaccines-12-00179-t001:** Crude and age-standardized all-cause mortality rates (MR) in Greenland, and crude and age-standardized mortality 31 days after invasive pneumococcal disease (IPD) hospitalization in Greenland for the total period (1995–2020), pre (1995–2010), and post (2010–2020) introduction of the 13-valent pneumococcal conjugate vaccine (PCV13) in Greenland.

Period	Population	Observed Deaths	PYRS	World Weighted ^1^ Deaths	World Weighted ^1^ PYRS	MR (95% CI)	MR (95% CI)
						Observed ^2^	World Weighted ^1,2^
Total Period (1995–2020)	Total, Greenland (1995–2020)	12,189	1,309,007	19,058.1	1,309,007	9.3 (9.1–9.5)	14.6 (14.4–14.8)
Pre-PCV13 Period (1995–2010)	Total, Greenland (1995–2020)	7336	809,145	13,233.4	809,145	9.1 (8.9–9.3)	16.4 (16.1–16.6)
Post-PCV13 Period (2010–2020)	Total, Greenland (1995–2020)	5363	550,399	7062.6	550,399	9.7 (9.5–10.0)	12.8 (12.5–13.1)
Total Period (1995–2020)	IPD patients, Greenland (1995–2020)	66	20.1	61.5	20.1	3292 (2546–4188)	3068 (2349–3936)
Pre-PCV13 Period (1995–2010)	IPD patients, Greenland (1995–2020)	49	13.7	45.7	13.7	3589 (2655–4745)	3347 (2448–4469)
Post-PCV13 Period (2010–2020)	IPD patients, Greenland (1995–2020)	17	6.4	28.4	6.4	2658 (1548–4255)	4441 (2961–6402)

Abbreviations: IPD: invasive pneumococcal disease, PCV13: 13-valent pneumococcal conjugate vaccine, PYRS: person-years, MR: mortality rate, CI: confidence interval. ^1^ Weighted with proportion of persons in the corresponding age groups of the World Health Organization (WHO) World Population Standard. ^2^ Per 1000 person-years.

**Table 2 vaccines-12-00179-t002:** Standardized mortality ratio (SMR) in patients with invasive pneumococcal disease (IPD) per 1000 person-years in Greenland for the total period (1995–2020), pre- (1995–2010), and post- (2010–2020) introduction of the 13-valent pneumococcal conjugate vaccine (PCV13) to the childhood vaccination program in 2010 in Greenland.

	Total Period (1995–2020)	Pre-PCV13 Period (1995–2010)	Post-PCV13 Period (2010–2020)	MRR (95% CI) ^1^	*p*-Value ^2^
Level	SMR (95% CI)	SMR (95% CI)	SMR (95% CI)		
Total	198.8 (156.2–253.1)	267.6 (202.2–354.1)	114.2 (71.0–183.7)	0.43 (0.25–0.74)	**0.003**
**Sex ^3^**					
Female	191.7 (125.0–294.0)	238.4 (150.2–378.4)	88.1 (28.4–273.3)	0.37 (0.11–1.26)	0.111
Male	202.3 (151.1–271.0)	288.1 (202.6–409.7)	121.9 (72.2–205.9)	0.42 (0.23–0.80)	**0.008**
**Age group**					
<5	443.8 (166.6–1182.5)	502.4 (188.5–1338.6)	0	0	-
5–39	435.5 (195.6–969.3)	557.7 (250.5–1241.3)	0	0	-
40–59	518.5 (372.2–722.1)	563.3 (391.5–810.7)	374.3 (168.2–833.2)	0.66 (0.28–1.60)	0.362
≥60	86.9 (56.7–133.3)	88.6 (47.7–164.6)	85.4 (47.3–154.3)	0.96 (0.41–2.27)	0.935
**Region ^4,5^**					
Nuuk	194.4 (135.9–278.0)	355.8 (240.4–526.6)	59.5 (24.8–142.9)	0.17 (0.06–0.44)	**<0.001**
North	539.9 (242.6–1201.9)	761.1 (316.8–1828.6)	220.1 (31.0–1562.8)	0.29 (0.03–2.48)	0.258
South	672.7 (372.6–1214.8)	703.9 (352.0–1407.5)	601.7 (194.1–1865.7)	0.85 (0.23–3.22)	0.817
East	145.4 (54.6–387.3)	103.8 (26.0–415.2)	242.3 (60.6–968.7)	2.33 (0.33–16.56)	0.397
West	204.6 (123.3–339.3)	186.2 (96.9–357.8)	240.2 (107.9–534.7)	1.29 (0.46–3.63)	0.629
**Ethnicity ^6^**					
Inuit	200.9 (157.3–256.7)	259.9 (195.3–345.9)	123.5 (76.8–198.6)	0.48 (0.27–0.83)	**0.009**
Non-Inuit	113.6 (28.4–454.3)	464.1 (116.1–1855.9)	0	0	-
**Charlson Comorbidity Index**					
Low (0)	285.0 (219.3–370.3)	355.6 (262.8–481.2)	178.5 (105.7–301.5)	0.50 (0.27–0.92)	**0.026**
Moderate (1–2)	44.7 (18.6–107.4)	75.7 (28.4–201.6)	17.0 (2.4–120.4)	0.22 (0.03–2.01)	0.181
High (<3)	211.5 (88.0–508.1)	246.7 (79.6–764.8)	174.2 (43.6–696.6)	0.71 (0.12–4.23)	0.703
***S. pneumoniae* serotypes**					
Vaccine type (VT) ^7^	110.6 (62.8–194.7)	112.0 (60.2–208.1)	104.2 (26.1–416.6)	0.93 (0.20–4.25)	0.926
Non-vaccine type (NVT)	135.0 (87.1–209.2)	303.4 (163.3–563.9)	86.8 (46.7–161.3)	0.29 (0.12–0.69)	**0.005**
Not serotyped	503.3 (349.8–724.3)	531.5 (359.2–786.6)	377.9 (141.8–1006.9)	0.71 (0.25–2.04)	0.527
Not isolated	283.2 (117.9–680.3)	289.6 (108.7–771.7)	259.9 (36.6–1845.4)	0.90 (0.10–8.03)	0.923

Abbreviations: PCV13: 13-valent pneumococcal conjugate vaccine, SMR: standardized mortality rate, CI: confidence interval, MMR: mortality rate ratio. North: towns of Qaanaaq, Upernavik, Uummannaq, South: towns of Paamiut, Qaqortoq, Narsaq, Nanortalik, East: towns of Tasiilaq, Ittoqqortoomiit, West: towns of Ilulissat, Qasigiannguit, Qeqertasuaq, Aasiaat, Kangaatsiaq, Sisimiut, Maniitsoq. ^1^ Ratios between standardized mortality rates for the post-PCV13 period compared with the pre-PCV13 period. ^2^ *p*-value based on Pearson’s chi-square test for statistical difference between the pre-PCV13 and the post-PCV13 periods. Bold values denote statistical significance at the *p* < 0.05 level. ^3^ One was unknown. ^4^ Nine were unknown. ^5^ Test of homogeneity of variance test: *p*-value 0.006. ^6^ Two were unknown. ^7^ PCV13 contains 13 serotypes of *S. pneumoniae*: 1, 3, 4, 5, 6A, 6B, 7F, 9V, 14, 18C, 19A, and 23F.

## Data Availability

Data on the population of Greenland are available PxWeb—Select table (stat.gl) https://bank.stat.gl/pxweb/en/Greenland/ (accessed on 5 November 2020). Data on the world population are available https://population.un.org/wpp/Download/Standard/Population/ (accessed on 10 August 2021). Microbiological data were acquired from the Microbiological Laboratory at the QIH, and from the Department of Microbiology and Infection Control at Statens Serum Institut, but are not publicly available. Data on comorbidity, course of hospitalization, and discharge date were acquired from the National Inpatient Registry in Greenland and are not publicly available. Data on death date for the study population were acquired from statistics Greenland and are not publicly available.

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
