# Peer review of "Mortality of Invasive Pneumococcal Disease following Introduction of the 13-Valent Pneumococcal Conjugate Vaccine in Greenland"

_vaccines, 2024, doi:10.3390/vaccines12020179_

Round 1
Reviewer 1 Report (Previous Reviewer 1)
Comments and Suggestions for Authors
Estimated Authors,
I've appreciated the considerable efforts you paid in improving your report on the mortality of IPD in Greenland.
I've noticed that you've addressed all my previous concerns, and those from Rev.2 as well. These interventions have improved the overall quality of the paper without eliciting any potential further shortcoming.
Therefore, I've no further requests and I'm endorsing the acceptance of the paper.
Author Response
Dear peer-reviewer,
thank you very much for your positive comments.
We are most happy that with our publication we will fill a gap in the exicting evidence on the subject.
Best wishes,
Kristiana Nikolova and co-authors
Reviewer 2 Report (Previous Reviewer 2)
Comments and Suggestions for Authors
I only have two minor comments:
Lines 269-270. Not only in the oldest age group (≥ 65 years); the decrease in the SMR in the age group 40-59 is also non-significant (0.362). Please, consider adding this data in the sentence.
Line 320. “…. there is no effect of PCV13 introduction on mortality of the elderly above 60 years of age.”
I recommend adding: “…. there is no effect of PCV13 introduction for children on the mortality of the elderly above 60 years of age.”
Author Response
Dear peer-reviewer,
thank you for your thorough comments on our article.
Our replies follow here:
Comment 1: We agree. We have deleted "(...) except for the oldest age group (≥ 65 years) where SMR remained virtually unchanged over the two periods."
Comment 2: We have inserted "children" in the sentence as you have recommended.
Best wishes,
Kristiana Nikolova and co-authors
This manuscript is a resubmission of an earlier submission. The following is a list of the peer review reports and author responses from that submission.
Round 1
Reviewer 1 Report
Comments and Suggestions for Authors
Estimated Authors,
Estimated Editors in Vaccines,
this is a very interesting study on the occurrence of pneumococcal infections and IPV in Inuit people before and after the introduction of PCV in Groenland.
Briefly, as expected accordingly to previously available international reports, PCV had positive effect on the occurrence of IPV and pneumoccal infection, and the protective effect was not limited to vaccinal serotypes, but somehow some positive effect was identified also on non-vaccinal serotypes.
The paper is well documented and written, and no specific reworking over main results is required, at least from my point of view.
On the contrary, I've some concerns about the following issues I've noticed.
1) the English text is substantially appropriate for the aims of this report, but some sentences may require a double check. For example: "IPD-related mortality was defined as related to IPD if death occurred within 31 days after IPD hospitalisation. If death did not occur, patients were censored 31 days after IPD hospitalisation" is grammarly correct, but it could run far better.
2) Please double check Table 1. Column 7+8 on the subsection about IPD suggest that (e.g.) MR (95%CI) for IPD patients would be 3292 (2546 to 4188) per 1000 person years. Could you please double check whether reporting is not affected by typos or not?
3) From my point of view, Table 2, section on age groups, is very important: according to your estimates, no death has occurred in post-PCV period in age group < 5 and 5 - 39! This is a common, highly expected but also significant result that should be stressed across the main text.
Comments on the Quality of English Language
see above
Author Response
please find the attached responses

Reviewer 2 Report
Comments and Suggestions for Authors
In general, the article is well written, the results are sound and support the conclusion drawn by the authors.
My main concern has to do with the comorbidities of dead patients. As IPD mortality is associated with older age and comorbidities, an analysis of the possible differences between the comorbidities of patients with IPD in both periods should be done to rule out a bias consequence of different predisposing conditions in the population of each period.
Also, the calculation of the standardized mortality rate (SMR) is not clear. If SMR is the ratio of observed to expected deaths in a population in a given period, I don’t see why SMR in patients with IPD is referred to the expected deaths in the general population (line 144) and not to the expected deaths due to IPD in the standard population. It would be also important to show the values of expected deaths in the general population and the calculation of Observed PYRS in deaths due to IPD (Table 1) should be better explained.
Minor comments
1. Line 51, 62, 196, … please, put the name Streptococcus pneumoniae in italics.
2. Was PCV7 used in Greenland on a private or public basis? Please, add this information in the introduction.
3. Lines 132-133. “Crude IPD .. and 1,000 PYRS as denominator.” 1000 PYRS of what population?
4. Line 168. Please, add the range or the SD to the average of days of hospitalization.
5. Do you have any information about the comorbidities of admitted patients in both periods? An analysis of the comorbidities between both periods would strength the conclusions of this work.
6. Lines 176. Please, include the p-value. If a significant difference was observed in the mortality rates, to say that there were “no major change” is not correct.
7. Line 179. The decease after age-standardization was significant? Please add the p-value.
8. Table 2. I suggest adding a column (before total period SMR) with the absolute number of isolates in each case to have a better perception of the incidences in the comparison with other studies. Please, add the serotypes included in the PCV13 as footnote.
9. Lines 245-246. Please, show how many isolates were collected in each period to show that there was no difference in the % of isolates serotyped by period. What was the % of isolates serotyped of dead patients?
10. Line 246. “…115 were serotype during the post-PCV13 period”. Shouldn´t be pre-PCV13 period?
11. Discussion is too long. May be it could be shortened (for instance, lines 261-270 are results…).
12. Lines 283-287. The discrepancies between crude and standardized data are quite confusing. In my opinion the message should be clearer: has the mortality increased or decrease after PCV13 vaccination of children?
13. Line 287. The number of IPD cases in each age group is not described in the results.
14. Lines 315. What are the “indirect effect” on unvaccinated adults? Please, be more precise (may be replace “effect” with “herd protection”).
15. Lines 316-317. The switch of serotypes in the elderly is not shown in the results. This sentence is speculative.
16. Lines 326-330. The relation and analysis of serotypes and comorbidities is not shown in the results. Thia material could be included as a Supplementary table.
17. Line 363. “The PCV13 may exert its effect… by reducing the severity of IPD”. However, in this study, all patients died in the second period were >40 years old, and so, had not received the PCV13. Therefore, this argument in this study could not be demonstrated. In my opinion, this paragraph (lines 358-371) is somehow redundant and could be deleted.
18. Lines 372-377. Nothing is said about the antibiotic resistance of isolates causing IPD. Isolates from the second period could have been more susceptible, in turn treatments more effective and in consequence mortality reduced.
Author Response
please find the attached responses
